# Inter-Day Variation in the Fasting Plasma Lipopolysaccharide Concentration in the Morning Is Associated with Inter-Day Variation in Appetite in Japanese Males: A Short-Term Cohort Study

**DOI:** 10.3390/metabo13030395

**Published:** 2023-03-08

**Authors:** Nobuo Fuke, Yusuke Ushida, Ikuo Sato, Hiroyuki Suganuma

**Affiliations:** 1Innovation Division, KAGOME Co., Ltd., 17 Nishitomiyama, Nasushiobara 329-2762, Tochigi, Japan; 2Department of Obstetrics and Gynecology, International University of Health and Welfare Hospital, Otawara 329-2763, Tochigi, Japan

**Keywords:** lipopolysaccharide, endotoxin, metabolic endotoxemia, appetite, carbohydrate intake, fat burning, breath acetone, body mass index

## Abstract

Injection of lipopolysaccharide (LPS), a product of gut bacteria, into the blood increases blood triglycerides and cortisol, an appetite-stimulating hormone. Meanwhile, small amounts of LPS derived from gut bacteria are thought to enter the bloodstream from the gut in daily basis. This study aimed to investigate the effect of LPS influx on appetite or lipid metabolism in humans in everyday life. We measured the fasting plasma LPS concentration before breakfast and the corresponding days’ appetite and fat-burning markers for 10 days in four Japanese males (28–31 years) and analyzed the correlation of their inter-day variation. The LPS concentration was negatively correlated with fullness, and positively correlated with the carbohydrate intake. Against our hypothesis, the LPS concentration was positively correlated with the fasting breath acetone concentration, a fat-burning marker. There was a positive correlation between the LPS concentration and fasting body mass index (BMI), but the inter-day variation in BMI was slight. The results suggest that the LPS influx in everyday life is at least associated with appetite in the day.

## 1. Introduction

Increased appetite [1] and the suppression of lipid metabolism [2] are important factors for obesity and metabolic syndrome. It has been demonstrated that metabolic syndrome leads to vascular disease and type 2 diabetes [3] and is a serious health issue worldwide. In addition, the cost of metabolic syndrome, including the cost of healthcare and the potential loss of economic activity, has been estimated to be about in trillions [4], making its prevention and alleviation an urgent economic matter as well. Because suppressing appetite, promoting fat-burning, and improving energy balance are key to preventing and alleviating obesity and metabolic syndrome, there is a long history of research on their mechanisms. Hormones, neuropeptides, and neurotransmitters are known to be the primary elements involved in the suppression of appetite in humans [5]. Extensive research has been performed on systems that regulate fat-burning based on the allosteric control, phosphorylation control, and expression control of key enzymes [6]. While progress toward an understanding of such internal control networks with regard to the control of appetite and fat-burning has been made, the effect of external factors, particularly the intestinal bacteria, has not yet been sufficiently described.

Lipopolysaccharide (LPS) is a molecule that is composed of lipids and polysaccharides and is produced by Gram-negative bacteria. A small amount of LPS, derived from gut bacteria, enters the bloodstream on a daily basis in association with dietary habits [7], and is thought to trigger a toll-like receptor 4-mediated inflammatory response [8]. It has been reported that this flow of LPS into the bloodstream may contribute to the onset and/or progression of obesity [9], insulin resistance [10], diabetes [11], nonalcoholic fatty liver disease [12], pancreatitis [13], amyotrophic lateral sclerosis [14], and Alzheimer’s disease [14]. In addition, previous reports suggest that LPS may be associated with appetite and lipid metabolism. Regarding appetite, an in vitro study [15] reported that LPS acts directly on the cells of the adrenal glands to promote cortisol secretion, and it has been reported in human experiments that LPS administered into the bloodstream increases the blood cortisol concentration [16,17]. As cortisol has been suggested to have appetite-promoting effects [18,19], it is likely that the flow of LPS into the bloodstream stimulates an increase in appetite in humans through cortisol secretion. Regarding lipid metabolism, an in vitro experiment using rabbit hepatocytes reported that LPS promotes triglyceride synthesis [20]. However, the effects of the flow of LPS into the bloodstream in everyday life on appetite and lipid metabolism in humans have not yet been elucidated.

In experiments in which LPS was administered into human blood, it was reported that the blood triglyceride and cortisol concentrations increased to the peak concentrations just 4–5 h following the administration and that they decreased back to the pre-administration levels at 12 h following the administration [21]. Changes in the blood LPS concentration are therefore conjectured to influence fat burning and appetite quickly and dynamically. Because of this, there are concerns that the relationship of the blood LPS concentration with fat burning and appetite cannot be captured in general prospective cohort studies that examine the relationship between the exposure to a given substance and changes in participants’ health statuses over many years. One previous study reported that the blood LPS concentration increases significantly after just 3 days of consuming a diet high in fructose [22]. Owing to the variation from day to day in what individuals eat, the blood LPS concentration is conjectured to exhibit intraindividual inter-day variation. That is, taking daily measurements of the blood LPS concentration and fat burning/appetite and evaluating the correlations in their variation may be one way to verify the relationship between the two. In fact, considering the aforementioned reports that the blood LPS concentration steadily affects the blood triglyceride and cortisol concentrations over 4–5 h [21] and varies throughout the day due to meals [12,23,24,25], measuring the fasting blood LPS concentration and evaluating fat-burning and appetite markers, either at the same time or subsequently throughout the day, would be an appropriate way to assess this relationship.

To that end, we conducted this study to determine whether the flow of LPS into the bloodstream in everyday life is a factor in the suppression of fat burning and/or in the enhancement of appetite by measuring (1) the fasting plasma LPS concentration and (2) fat-burning and appetite markers in Japanese males throughout each day over a period of 10 days and by evaluating the correlations in the inter-day variation among those measurements. The fasting plasma triglyceride (TG) concentration [26], fasting breath acetone concentration [27,28], and fasting body mass index (BMI) were used as the fat-burning markers, whereas the appetite scores before and after each meal [29] and nutrient intake at each meal were used as the appetite markers.

## 2. Materials and Methods

### 2.1. Ethics Approval and Consent to Participate

This study was conducted in accordance with the guidelines of the 2013 version of the Declaration of Helsinki, and all procedures involving human subjects were approved by the ethics committees of KAGOME CO., LTD. (2020-R13). Although the participants in this study were employees of the same company as the researchers, during the study briefing it was explained, both verbally and in writing, that participation or withdrawal from the study was voluntary and that there were no disadvantages for not participating or withdrawing from the study. Written informed consent was obtained from all participants.

### 2.2. Participants

The study participants were recruited through an email that we sent to the employees of KAGOME CO., LTD. Those who satisfied the requirements for participating in this study, did not violate the exclusion criteria, and were considered appropriate by the doctor responsible for the experiment and the principal investigator were chosen as participants.

The requirements for participation were as follows: (1) male, (2) aged between 20 and 39 years, inclusive, (3) BMI of at least 22 kg/m^2^, (4) consumes ≤240 g of vegetables on at least 1 day during the week, (5) feels that he gains weight more easily than others, and (6) feels he cannot control his appetite at least 1 day each week.

We chose males as participants (condition 1) because it has been reported that the menstrual period affects appetite in females [30], and we were concerned that this would affect the evaluation of the relationship between the fasting plasma LPS concentration and appetite. We chose individuals younger than 40 years as participants (condition 2) to control for the influence of aging-related changes, based on the fact that males (in addition to females) have been reported to undergo changes in hormone balance in their forties and onward [31]. We chose individuals with a BMI exceeding the 22 kg/m^2^ level (condition 3), that the Ministry of Health, Labor, and Welfare defines as appropriate for Japanese people, because blood LPS concentration positively correlates with BMI [32,33] thus individuals with a low BMI were concerned to exhibit a plasma LPS level below the detectable limit.

As the intake of vegetables may suppress the flow of LPS into the bloodstream [7], it would also be likely that the fasting plasma LPS concentrations of individuals with a varying intake level of vegetables would fluctuate, and we thus supposed such individuals would be appropriate for the assessment in this study. Accordingly, we stipulated that the participants must have a lower vegetable intake on at least 1 day each week (condition 4). The mean vegetable intake of Japanese people is 280.5 g [34]. We told the applicants that one handful of vegetables is 60 g so that they could easily assess their own vegetable intake, with four handfuls (i.e., 240 g, which is less than the mean vegetable intake of Japanese people) serving as the reference value. Conditions 5 and 6 were added because individuals with large variability in fat burning and appetite were thought to be appropriate for this study.

The exclusion criteria were as follows: (1) exhibiting factors other than their eating habits that would greatly affect their blood LPS concentration (specifically, having an infectious disease, a chronic underlying condition, or a habit of smoking or drinking); (2) exhibiting factors other than the blood LPS concentration that would greatly affect their appetite (specifically, skipping meals or undergoing hormone therapy); (3) having an adverse reaction to having their blood drawn (specifically, being so oversensitive to alcohol that it would be difficult to use alcohol-based disinfectants, or having a skin disease on the fingertips); (4) participating in other human experiments in the month prior to the start of this experiment; and (5) being deemed unfit to participate for the experiment by the doctor responsible or the principal investigator.

### 2.3. Study Design

This study was conducted as a short-term cohort study from Sunday, 8 November 2020, to Friday, 20 November 2020. From Sunday through Friday during the two consecutive weeks of the study, we had the participants measure their own whole day physical activity levels and record the start time, end time, and duration of each meal, the nutrients they consumed at each meal. In addition, from Monday through Friday in each week, we had them collect fasting blood themselves, measure their own fasting BMI and fasting breath acetone concentration before breakfast, and measure their own appetite score at each meal. If they snacked or otherwise ate between meals, we had them record the start and end times, duration, and nutrients for those instances as well.

During the experimental period, the participants adhered to the following five restrictions: (1) leave at least 8 h between the last instance of eating or drinking each day and drawing blood the next day (only water was allowed) because the transient rise in the blood LPS concentration from meals takes approximately 5 h to return to baseline [12,23,24,25] and because the triglyceride concentration similarly takes approximately 8 h to return to baseline [35,36]; (2) maintain the same normal eating habits, sleep schedule, and exercise routine as much as possible while adhering to the restrictions; (3) from Monday to Friday, do not skip meals, force yourself to eat, or engage in unusually intense exercise, as it is assumed that these influence the evaluation of appetite and fat burning; (4) do not drink (alcohol) or smoke, as there is a risk this may influence the measurements of the fasting plasma LPS concentration and breath acetone concentration; and (5) as much as possible, avoid taking medicines or supplements other than those you take regularly, and log them if you do.

To protect the participants’ personal information and avoid evaluation bias, in this study, the delivery and collection of measurement devices, blood samples, and logs were performed in such a way that did not involve meeting in person. For the same reason, the results of the experiment were not disclosed to anyone other than the researchers until all analyses were completed and the data were fixed.

### 2.4. Self-Collection of Blood from Fingertips

LPS from the living environment adheres to one’s fingertips and must be removed before blood is drawn from the fingertips. When we conducted preliminary tests of the LPS-eliminating function of cotton swabs impregnated with 10–90% (*v/v*) ethanol, we observed that wiping the fingertips five or more times with a 50% (*v/v*) ethanol-impregnated swab was the most effective method for eliminating LPS (Appendix A). As such, we asked the participants in this study to wipe their fingertips five or more times with a 50% (*v/v*) ethanol-impregnated swab. To check for contamination by LPS originating from the skin of the fingertips, we had the participants apply 100 μL LPS-free distilled water (Otsuka distilled water; Otsuka Pharmaceutical Co., Ltd., Tokyo, Japan) to their fingertip with an eye dropper and collect it in a heparin tube (Microvette 100 LH; Sarstedt AG & Co. KG, Nümbrecht, Germany). Then, they pricked their fingertip with a lancet (Medisafe-Fine Touch; Terumo Corporation, Tokyo, Japan) and collected approximately 100 μL of the drawn blood, similarly, in a separate heparin tube. The collected LPS-free distilled water and blood were immediately placed in a container with a refrigerant and submitted to the researcher in the morning. The blood was centrifuged at 1200× *g* for 15 min at 4 °C to prepare the plasma. A portion of the plasma was submitted for the LPS concentration measurement by the end of the day, whereas the remaining plasma was kept at −80 °C in cold storage for the measurement of triglyceride concentration.

### 2.5. Measurement of Plasma LPS Concentration

The plasma samples were diluted 10-fold with LPS-free distilled water, then heated at 70 °C for 10 min. Then, 50 μL of each sample and limulus amebocyte lysate (LAL) reagent (Endospecy; Seikagaku Corporation, Tokyo, Japan) were mixed in a 96-well plate (AGC Techno Glass Co., Ltd., Shizuoka, Japan), and the plate was incubated at 37 °C for 60 min (plate #1). The LPS-free water collected from the fingertips was handled in the same fashion as the plasma. To correct for the absorbance (Abs) originating from the color of the plasma, 50 μL of the plasma and LPS-free distilled water were mixed in a 96-well plate and incubated at 37 °C for 60 min (plate #2). After the incubation, the Abs of the sample was measured at 405 nm (reference Abs: 492 nm) using a microplate reader (Corona Electric Co., Ltd., Ibaraki, Japan). The Abs originating from the LAL reaction was calculated as follows:

(Abs of ‘plasma + LAL reagent’ well [plate #1]) − (Abs of blank well [plate #1]) − ([Abs of ‘plasma + water’ well {plate #2}] − [Abs of blank well {plate #2}])

Control standard endotoxin (Seikagaku Corporation), in the range of 0.0001, 0.0004, 0.0016, 0.0063, and 0.0250 endotoxin unit (EU)/mL, was reacted with the LAL reagent in the same way as the plasma in plate #1, and a calibration curve was drawn.

### 2.6. Measurement of Plasma TG Concentration

The plasma frozen at −80 °C was thawed and submitted for the measurement of the TG concentration. Measurements were performed using a commercially available kit (triglyceride quantification kit; Cell Biolabs, Inc., San Diego, CA, USA) according to the manufacturer’s instructions.

### 2.7. Measurement of Breath Acetone Concentration

The participants themselves used a portable breath acetone analyzer (Ketonix; Ketonix AB, Varberg, Sweden) and dedicated smartphone app to measure their own breath acetone concentration. They logged the resulting measurements and submitted them to the researchers.

### 2.8. Measurement of BMI

The participants weighed themselves with their scale at home and calculated their BMI by dividing their weight (kg) by the square of their height (m). They logged the resulting measurements and submitted them to the researchers.

### 2.9. Measurement of Appetite Scores

The appetite scores were assessed according to the previous report [29]. In brief, the participants self-assessed their desire to eat, hunger, fullness, and prospective food consumption (PFC) on a 150-mm visual analog scale before every meal and every 10 min from 0 to 60 min after every meal (breakfast, lunch, and dinner). They logged the resulting assessments and submitted them to the researchers.

### 2.10. Measurement of Nutrient Intake

The participants self-assessed their nutrient intake for each meal and snack using a smartphone dish-based dietary records app called “asken” [37]. The nutrients they assessed were calories, fat, protein, carbohydrates, dietary fiber, calcium, iron, vitamins (A, E, B_1_, B_2_, and C), saturated fat, and sodium. They logged the resulting measurements and submitted them to the researchers.

### 2.11. Measurement of Physical Activity Level

The participants went about their daily life with an activity tracker (HJA-750C; Omron Corporation, Kyoto, Japan) fastened to the waistband of their pants, except when they were bathing or sleeping. After the experiment was finished, the researchers collected the activity trackers in such a way that did not involve meeting in person and used specialized software (version 2.2; Omron Corporation) to extract and evaluate the recorded physical activity levels, which were reported in metabolic equivalents (METs).

### 2.12. Statistical Analysis

To assess how the analysis would be affected by LPS originating from the skin of the fingertips that had contaminated in the blood, we used Spearman’s rank correlation coefficient to evaluate the correlation between (1) the fasting plasma LPS concentration and (2) the fasting plasma LPS concentration minus the LPS concentration of the LPS-free distilled water that was applied to the skin. We performed a Smirnov–Grubbs test on each individual fasting plasma LPS concentration and considered those with *p* < 0.05 to be outliers. If any of the fasting plasma LPS concentration data were considered to be an outlier, that data and other measurements obtained on the same day were excluded from the subsequent analyses.

From the remaining dataset, we took the data from the first day we could obtain the fasting plasma LPS concentration for each participant, used it as that participant’s baseline, and calculated each measurement’s change from baseline for each day. What is labeled later in this paper as the “change in X” refers to this change from baseline. We pooled together these calculated change data for all participants to analyze the relationship between the change in fasting LPS plasma concentration and changes in fat-burning and appetite markers. We calculated the appetite scores by calculating the area under the curve (AUC) for each meal from before eating until 60 min after eating, which we then submitted for the statistical analysis.

To perform a multivariate analysis on the measured values, we used a generalized linear mixed model (GLMM), including subjects-level random effects [38]. When evaluating the relationship between the fasting plasma LPS concentration and fat-burning markers, we used the change in fasting plasma TG concentration, fasting breath acetone concentration, or fasting BMI as the objective variable; the change in fasting plasma LPS concentration as the explanatory variable; and the change in duration of fasting since the previous night as an adjustment factor. When evaluating the effect of the fasting plasma TG concentration on the relationship between the fasting breath acetone concentration and fasting plasma LPS concentration, we used the change in fasting plasma TG concentration as an adjustment factor.

When evaluating the relationship between the fasting plasma LPS concentration and appetite markers, we used the change in appetite score as the objective variable; the change in fasting plasma LPS concentration as the explanatory variable; and the change in calorie intake for the meal in question, the change in duration of the meal in question, the change in calorie intake for the previous meal, and the change in level of physical activity from the previous meal to the meal in question as adjustment factors. Finally, for each nutrient, we used the change in intake at each meal or the change in its total daily intake as the objective variable; the change in fasting plasma LPS concentration as the explanatory variable; and the change in preprandial hunger score for each meal and the change in day’s total physical activity level as adjustment factors.

In the GLMM, *t* > 2 and *t* < −2 were considered to be significant. The statistical analysis was performed using R statistical package (version 4.0.3; R Foundation for Statistical Computing, Vienna, Austria) and EZR (version 1.40), an easy-to-use software created by Kanda et al. based on R [39].

## 3. Results

### 3.1. Participant Characteristics

It was estimated that there were approximately 30 potential study participants who met the inclusion criteria of age, gender and BMI. After recruitment, twelve people participated in the study briefing. Of these, three withdrew for personal reasons. A further five were excluded because they met the exclusion criteria. Therefore, the study was conducted with the remaining four participants. Their ages ranged from 28–31 years, their BMIs when the experiment started ranged from 22.3–25.1 kg/m^2^, there were 1–3 days per week when their vegetable intake was 240 g or less, and there were 1–3 days per week when they could not control their appetite. Participants in this study did not take any medications or dietary supplements during the study period. The fasting plasma LPS concentration of each participant (labeled #A, #B, #C, and #D) is shown in Figure 1A. The mean, standard deviation (SD), and coefficient of variation (CV) of these data over 10 days for each participant was as follows. #A: 0.045 ± 0.030 EU/mL (CV = 66%); #B: 0.018 ± 0.023 EU/mL (CV = 131%); #C: 0.018 ± 0.014 EU/mL (CV = 80%), and #D: 0.010 ± 0.003 EU/mL (CV = 26%). The detection limit for plasma LPS concentrations in this study was 0.0001 EU/mL. The lowest plasma LPS concentrations for each participant (#A, #B, #C and #D) were 0.0163, 0.0032, 0.0077 and 0.0048 EU/mL, respectively, all above the detection limit. The LPS concentration of the LPS-free distilled water that was used to check for LPS contamination from the skin was low relative to the fasting plasma LPS concentration (Figure 1B). Because a significant positive correlation was demonstrated between the fasting plasma LPS concentration and fasting plasma LPS concentration minus the LPS concentration of the LPS-free distilled water that was applied to the skin (Spearman’s rank correlation coefficient analysis; ρ = 0.91, *p* = 6 × 10^−16^), the degree of LPS contamination from the skin that occurred when blood was drawn was determined to be negligible. Accordingly, the actual fasting plasma LPS concentration measurements were used directly in the subsequent analyses.

As Figure 1 exhibits, the participants’ fasting plasma LPS concentrations exhibited considerable variation over the 10 days. In particular, #B and #C had high values only at the start of the experiment, and we therefore considered day 1 for #B and days 1 and 2 for #C to be outliers based on the Smirnov–Grubbs tests (*p <* 0.05). These data were excluded from the subsequent analyses because outliers may cause the data to deviate from the participants’ normal variation in the fasting plasma LPS concentration. This resulted in a final total number of n = 37 data points pooled from the four participants.

### 3.2. Correlation between Plasma LPS Concentration and Fat-Burning Markers

The change in fasting plasma LPS concentration exhibited a significant positive correlation with the changes in breath acetone concentration and change in BMI (Table 1). This correlation was maintained even after adjusting for the change in duration of fasting since the previous night. The breath acetone concentration is a marker of fat burning and is thought to be associated with the plasma TG concentration. As such, we performed a GLMM analysis on it with the change in fasting breath acetone concentration as the objective variable, change in fasting plasma LPS concentration as the explanatory variable, and changes in duration of fasting and in fasting plasma TG concentration as adjustment factors. The significant positive correlation between the fasting breath acetone concentration and fasting plasma LPS concentration was maintained in this analysis as well (*t* = 7.7). Furthermore, the fasting breath acetone concentration exhibited a significant negative correlation with the fasting plasma TG concentration in this analysis (*t* = −2.1).

### 3.3. Correlation between Plasma LPS Concentration and Appetite Scores

The change in fasting plasma LPS concentration exhibited a significant negative correlation with the change in AUC for fullness score at breakfast and lunch (Table 2). This correlation was maintained even after adjusting for the change in calorie intake at the meal in question (Model 2), the change in time spent eating the meal in question (Model 3), the change in calorie intake at the previous meal (Model 4), and the change in level of physical activity from the previous meal to the meal in question (Model 5).

### 3.4. Correlation between Plasma LPS Concentration and Nutrient Intake

We performed a GLMM analysis on the correlation between the change in fasting plasma LPS concentration and change in nutrient intake at each meal. The analysis demonstrated that, regardless of whether or not the preprandial hunger score was adjusted for, the change in fasting plasma LPS concentration exhibited a negative correlation with the changes in protein, calcium, vitamin B_1_, and salt intakes at breakfast and the changes in calcium and vitamin A intakes at lunch (Table 3). Furthermore, also regardless of whether or not the preprandial hunger score was adjusted for, the change in fasting plasma LPS concentration exhibited a significant positive correlation with the change in caloric, carbohydrate, and salt intakes at dinner and a significant negative correlation with the change in vitamin B_1_ intake. When we performed a GLMM analysis on the correlation between the change in fasting plasma LPS concentration and change in total daily intake of each nutrient, the change in fasting plasma LPS concentration demonstrated a significant positive correlation with the change in total daily carbohydrate intake, regardless of whether or not the preprandial hunger score was adjusted for (Table 4), and the results were similar when the nutrient intake from snacks was not included (Table 5).

## 4. Discussion

In this study, we measured the (1) the fasting plasma LPS concentration and (2) fat-burning and appetite markers in Japanese males throughout the day for 10 days and evaluated the correlations in the inter-day variation among those measurements to determine the relationship of the flow of LPS into the bloodstream in everyday life with fat-burning and appetite. The results demonstrated that, among the fat-burning markers, the change in fasting plasma LPS concentration exhibited a positive correlation with the change in fasting breath acetone concentration and in fasting BMI, whereas among the appetite-related markers, the fasting plasma LPS concentration exhibited a positive correlation with the change in the dinnertime and total daily carbohydrates intake and a negative correlation with the change in fullness.

As stated in the introduction, we hypothesized that the flow of LPS into the bloodstream suppresses fat burning. If this hypothesis is correct, the change in fasting plasma LPS concentration should exhibit a positive correlation with the change in fasting plasma TG concentration and change in fasting BMI and a negative correlation with the change in fasting breath acetone concentration. However, no correlation was observed between the change in fasting plasma LPS concentration and change in fasting plasma TG concentration. While the change in fasting plasma LPS concentration did exhibit a significant positive correlation with the change in fasting BMI, the change in fasting BMI of each participant in this study was ±0.1 kg/m^2^. Because the mean height of Japanese adult males is 167.7 cm [34], 0.1 kg/m^2^, in terms of the BMI, is 0.3 kg of body weight. Because Japanese adults excrete 200–400 mL of urine at a time [40], this means that the participants’ BMI in this study changed by a trivial amount, an amount that is on the level of a single episode of urination. Accordingly, it is difficult to conclude from this study whether or not the correlation we observed between the change in fasting plasma LPS concentration and the change in fasting BMI has physiological significance.

In contrast, a significant positive correlation was observed between the change in fasting plasma LPS concentration and change in fasting breath acetone concentration, which is contrary to our hypothesis. By the same GLMM analysis we also observed a negative correlation between the change in fasting breath acetone concentration and the change in plasma TG concentration. These results are consistent with those of previous reports [27,28] that determined that the breath acetone concentration is a marker of fat burning, which suggests that inter-day variation in the fasting plasma LPS concentration is associated with variation in fat burning. However, the fat burning associated with the fasting plasma LPS concentration is independent of the fat burning reflected by the fasting plasma TG concentration. The fasting plasma TG concentration is thought to be the product of fatty acids secreted by adipose tissue after they are absorbed by the liver, where they are synthesized into TG, incorporated into very low-density lipoprotein (VLDL), and secreted into the bloodstream [41]. In this process, some fatty acids in liver cells undergo *β*-oxidation [41], thereby producing acetone and other ketones. As such, the plasma TG concentration is thought to reflect the degree of *β*-oxidation in this process. Meanwhile, it has been reported that almost no TG synthesized de novo in the liver are incorporated into VLDL [42], making it likely that such TG are governed by a different synthetic and metabolic pathway than the one that governs the TG that use fatty acids originating from adipose tissue. As stated in the introduction, it has been reported that LPS stimulates the de novo genesis of TG in liver cells [20], and for this reason, the fact that a positive correlation was observed between the change in fasting plasma LPS concentration and change in fasting breath the acetone concentration, independent of the change in fasting plasma TG concentration, in this study may mean that the flow of LPS into the bloodstream in everyday life affects the de novo metabolism of triglycerides in the liver.

This study demonstrated a negative correlation between the change in fasting plasma LPS concentration and change in fullness at breakfast and lunch and a significant positive correlation between the change in fasting plasma LPS concentration and changes in both dinnertime and total daily carbohydrate intake. This suggests that a high fasting plasma LPS concentration on a given day is associated with appetite on that day, as we hypothesized it would be. We could not identify the mechanism underlying that relationship in this study, but our results suggest the following possibilities: As stated above, a significant positive correlation was observed between the change in fasting plasma LPS concentration and change in fasting breath acetone concentration, which suggests the possibility that an increase in the former promotes fat burning. If it was simply the case that the flow of LPS into the bloodstream promoted fat burning, individuals with a high fasting plasma LPS concentration would likely have a low plasma TG concentration and low BMI. However, in practice, the opposite was true, and participants with a high absolute (i.e., not change in) fasting plasma LPS concentration also had a high BMI and high plasma TG concentration (Appendix A). This contradiction can be explained by lipotoxicity. Lipotoxicity is the phenomenon whereby fat accumulates ectopically in the liver or muscles rather than in adipose tissue, damaging those tissues and increasing their insulin resistance [43,44]. The fact that LPS can at least promote the de novo synthesis of TG in the liver is consistent with the findings of a previous report [20], and it has been reported in experiments on rats [45] and humans [46,47] that LPS increases insulin resistance. The breath acetone concentration is higher in individuals with diabetes than in healthy individuals and is correlated with the blood glucose and hemoglobin A1c levels and thus reflects a state of high insulin resistance—that is, a state in which cells use fat as a source of energy because they cannot use glucose from the blood [44,48,49]. Consequently, if it is the case that the flow of LPS into bloodstream leads to lipotoxicity and increases insulin resistance, then that would also conceivably cause simultaneous increases in the plasma TG, BMI, and breath acetone concentration. Insulin also functions as an appetite-suppressive hormone. Because the appetite of normal-weight individuals is suppressed by insulin but that of obese individuals is not [50], an increase in insulin resistance should impede appetite regulation. It has also been reported that insulin resistance is associated with the desire to consume carbohydrates [51,52,53]. Consequently, if the results of this study are interpreted in the context of insulin resistance, then one could conclude that having a high fasting plasma LPS concentration causes insulin resistance to increase, effectively making it difficult to achieve satiety and strengthening the desire to consume carbohydrates. In the future, it will be necessary to prove the relationship between the fasting plasma LPS concentration and insulin resistance in healthy individuals to better understand this mechanism.

Although not the main objective of this study, we would like to mention factors associated with the inter-day variation in plasma LPS levels observed in this study. Previous reports suggest that changes in the gut microbiota associated with changes in dietary habits occur within two days [54]. Therefore, it is possible that what the subjects ate during the study period (e.g., the day before the blood sampling) may have altered the gut microbiota, which may have affected the amount of LPS that entered the blood. Probiotics and prebiotics are well known as potential influences on blood LPS levels [7]. In our previous cross-sectional study, we have found dietary factors associated with plasma LPS-binding protein concentrations (LPS exposure indicator): vitamins, carotenoids, lipids, fatty acids, dietary fiber, potassium, alcohol, and several foods (fish, green tea, cruciferous vegetables, and tomato) [55]. Participants in the current study did not take any medications or dietary supplements during the study period, thus excluding the possibility that they influenced the inter-day variation in plasma LPS concentrations. We conducted preliminary analysis of the relationship between nutrient intake the previous day and plasma LPS concentrations the following morning, and a positive association between vitamin B_1_ intake at lunch the previous day and plasma LPS concentrations the following morning was observed (In GLMM with change in plasma LPS concentration as the objective variable, change in vitamin B1 intake at lunch on the previous day as the explanatory variable and change in plasma LPS concentration, fasting time and daily calorie intake on the previous day as adjustment factors; *β* = 0.03, *t* value = 2.3). However, no association was found between vitamin B_1_ intake at breakfast, dinner, and throughout the day and plasma LPS concentrations, thus the results were inconsistent. Daily intake of probiotics, prebiotics or other foods was not assessed in this study. The composition of the gut microbiota was also not assessed in this study. Foods contain a wide range of components in addition to nutrients, for example tomatoes have been suggested to contain 7118 components [56]. The inter-day variation in plasma LPS concentrations observed in this study may be related to the variety of components (other than major nutrients) provided by the foods consumed daily and the resulting changes in the gut microbiota. Further studies are expected to be conducted to reveal the underlying mechanisms.

Finally, we describe the limitations of this study. LPS is a molecule composed of lipid A and polysaccharides, but the structure of lipid A differs between bacteria, which is thought to have different effects on the organism [57]. In our study, we used the LAL test, which comprehensively assesses the number of LPS molecules and their ‘activity’, rather than simply measuring the number of LPS molecules, as in the enzyme-linked immunosorbent assay, in order to reduce the influence of differences in LPS origin (i.e., activity). The LAL test is an assay based on the activation of a group of enzymes in limulus amebocyte lysate by LPS, the results of which are known to be associated with pyrogenicity and have been used to diagnose sepsis in humans [58]. Therefore, we believe that the study design was able to reflect the biological effects of LPS to some extent, even if there were structural differences in LPS due to differences in the gut microbiota between individual participants. On the other hand, it has been reported that the activity of some forms of LPS aggregation can be detected in the LAL test, despite the absence of induction of inflammatory cytokines [57]. This study included four participants who were employees of the company and lived in the same area, Nasushiobara, Tochigi, Japan. Therefore, it is possible that the gut microflora was comparatively similar between these subjects. It is necessary to investigate whether an association between plasma LPS concentrations measured by the LAL test and appetite can be found in populations with completely different national, racial and dietary habits, i.e., where the composition of the gut microbiota is expected to be very different. In addition, due to the small scale of this study, the reproducibility of its results must be confirmed in a larger-scale cohort study. Also, in this study, we inferred a causal relationship between the fasting plasma LPS concentration and fat burning or appetite, both of which were based on time differences. To verify this causal relationship, an interventional study must be conducted in which appetite is assessed after LPS is administered into the bloodstream.

## 5. Conclusions

This study demonstrated that the fasting plasma LPS concentration exhibits inter-day variation and that this variation is associated with appetite throughout the day. The results suggest that LPS from gut bacteria, which enters the bloodstream in everyday life, may affect our appetite. Blood LPS has been almost exclusively examined by longitudinal studies and case-control studies that focus on its relationship with disease, with hardly any attention devoted to their short-term effects on physiological function. It is hoped that more research will be done in the future to find out how LPS in the blood affects fluctuations in our health as we go about our daily lives.

## Figures and Tables

**Figure 1 metabolites-13-00395-f001:**
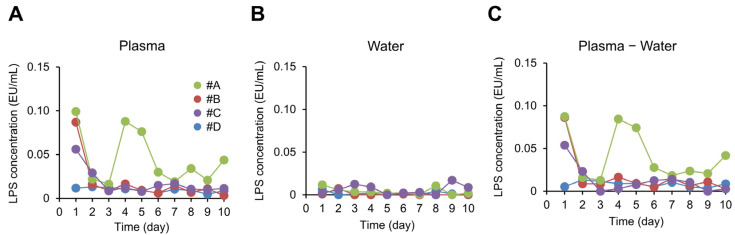
Inter-day variation in the fasting plasma lipopolysaccharide (LPS) concentration. Depicted are the fasting plasma LPS concentration (**A**), the LPS concentration in LPS-free distilled water that was applied to the fingertips (**B**), and the value of A minus B (**C**) for each of the 10 days for each of the four participants (participants #A through #D).

**Table 1 metabolites-13-00395-t001:** Relationship between the change in each fat-burning marker with the change in fasting plasma lipopolysaccharide concentration.

Variable	Model 1		Model 2	
	*β*	*t* Value	*β*	*t* Value
Change in fasting plasma TG concentration	−144	−0.6	−264	−1.2
Change in fasting breath acetone concentration	1146	7.8 *	1145	7.6 *
Change in fasting BMI	2	2.4 *	2	2.2 *

Generalized linear mixed model with * *t* > 2, with no adjustment factor (Model 1), or adjusted for the change in duration of fasting (Model 2). TG; triglyceride, BMI; body mass index.

**Table 2 metabolites-13-00395-t002:** Relationship between the change in the area under the curve for each appetite score and change in fasting plasma lipopolysaccharide concentration.

Variable		Model 1		Model 2		Model 3		Model 4		Model 5	
		*β*	*t* Value	*β*	*t* Value	*β*	*t* Value	*β*	*t* Value	*β*	*t* Value
Breakfast	Desire to eat	263	0.4	142	0.2	304	0.4	302	0.4	76	0.1
	Hunger	371	0.6	356	0.5	464	0.6	497	0.7	196	0.3
	Fullness	−2239	−4.2 *	−2426	−4.6 *	−2536	−4.0 *	−2769	−4.7 *	−2011	−2.8 *
	PFC	−220	−0.3	−596	−0.7	−207	−0.2	−238	−0.3	−715	−1.1
Lunch	Desire to eat	−438	−0.9	−434	−0.9	−440	−0.9	−532	−1.0	−880	−1.4
	Hunger	426	0.6	386	0.6	245	0.4	163	0.2	−110	−0.2
	Fullness	−3538	−3.2 *	−3828	−3.4 *	−3766	−3.4 *	−3766	−3.2 *	−3540	−3.3 *
	PFC	−115	−0.1	−88	−0.1	621	1.0	−50	−0.1	−1066	−1.4
Dinner	Desire to eat	−723	−1.1	−386	−0.6	−91	−0.1	38	0.1	310	0.5
	Hunger	−1095	−1.5	−680	−0.9	−313	−0.4	−234	−0.3	59	0.1
	Fullness	1060	1.3	336	0.4	−35	−0.1	−64	−0.1	−310	−0.4
	PFC	−1346	−1.3	−379	−0.4	81	0.1	207	0.2	363	0.4

Generalized linear mixed model with * *t* > 2 or *t* < −2, with no adjustment factor (Model 1) or adjusted for the change in calorie intake at the meal in question (Model 2). Model 3: Model 2 adjusted for the change in time spent eating the meal in question. Model 4: Model 3 adjusted for the change in calorie intake at the previous meal. Model 5: Model 4 adjusted for the change in level of physical activity from the previous meal to the meal in question. PFC; prospective food consumption.

**Table 3 metabolites-13-00395-t003:** Relationship between the change in nutrient intake at each meal and change in fasting plasma lipopolysaccharide concentration.

Variable	Breakfast				Lunch				Dinner			
	Model 1		Model 2		Model 1		Model 2		Model 1		Model 2	
	*β*	*t* Value	*β*	*t* Value	*β*	*t* Value	*β*	*t* Value	*β*	*t* Value	*β*	*t* Value
Calories	−1929	−1.5	−1937	−1.5	1197	0.9	492	0.4	3296	2.3 *	4096	3.4 *
Protein	−161	−2.9 *	−164	−3.0 *	95	1.1	49	0.6	225	1.7	282	2.1 *
Fat	−4	−0.1	−6	−0.1	84	0.7	43	0.3	−21	−0.2	−9	−0.1
Carbohydrate	−140	−1.0	−139	−1.1	47	0.3	68	0.4	648	4.2 *	723	3.8 *
Dietary fiber	−22	−1.1	−23	−1.1	1	0.1	5	0.2	−31	−0.9	−27	−0.8
Calcium	−2341	−2.7 *	−2444	−2.8 *	−871	−2.2 *	−1015	−2.3 *	−601	−0.4	382	0.3
Iron	−13	−1.6	−13	−1.6	5	0.7	3	0.4	12	0.6	22	1.1
Vitamin A	5157	1.7	6635	2.4 *	−4897	−3.3 *	−5545	−3.5 *	−1221	−0.7	−1314	−0.7
Vitamin E	−1	−0.1	−1	−0.1	−15	−0.9	−20	−1.1	17	1.1	21	1.2
Vitamin B_1_	−1	−2.2 *	−2	−3.0 *	2	1.0	1	0.7	−3	−2.7 *	−3	−2.1 *
Vitamin B_2_	−1	−1.1	−1	−1.1	0	−0.3	−1	−0.7	2	1.9	1	1.6
Vitamin C	554	1.5	647	1.8	−238	−1.1	−251	−1.1	−624	−1.9	−502	−1.4
Saturated fatty acids	1	0.0	2	0.1	−45	−1.6	−65	−2.1 *	−22	−0.7	−26	−0.8
Salt	−20	−2.2 *	−20	−2.1 *	17	1.2	15	0.9	42	2.7 *	48	2.9 *

Generalized linear mixed model with * *t* > 2 or *t* < −2, with no adjustment factor (Model 1), or adjusted for the change in hunger score before the corresponding meal (Model 2).

**Table 4 metabolites-13-00395-t004:** Relationship between the change in daily intake of each nutrient and change in fasting plasma LPS concentration.

Variable	Model 1		Model 2	
	*β*	*t* Value	*β*	*t* Value
Calories	2718	1.0	3107	0.9
Protein	122	0.6	62	0.3
Fat	61	0.3	100	0.5
Carbohydrate	524	2.4 *	579	2.3 *
Dietary fiber	−47	−0.8	−65	−0.9
Calcium	−4448	−2.1 *	−3374	−1.5
Iron	−3	−0.1	−1	0.0
Vitamin A	1565	0.4	2693	0.6
Vitamin E	−43	−2.1 *	−42	−1.8
Vitamin B_1_	−4	−1.5	−6	−2.8 *
Vitamin B_2_	−2	−1.0	−3	−1.8
Vitamin C	−362	−0.9	−196	−0.4
Saturated fatty acids	−30	−0.6	−19	−0.4
Salt	47	2.4 *	38	1.7

Generalized linear mixed model with * *t* > 2 or *t* < −2, with no adjustment factor (Model 1), or adjusted for the change in preprandial hunger score at breakfast, lunch, and dinner (Model 2).

**Table 5 metabolites-13-00395-t005:** Relationship between the change in daily intake of each nutrient and change in fasting plasma LPS concentration (excluding snacks).

Variable	Model 1		Model 2	
	*β*	*t* Value	*β*	*t* Value
Calories	3307	1.2	3795	1.2
Protein	180	0.9	137	0.6
Fat	99	0.5	145	0.7
Carbohydrate	501	2.5 *	543	2.4 *
Dietary fiber	−42	−0.7	−60	−0.9
Calcium	−4326	−2.0 *	−3143	−1.4
Iron	10	0.4	16	0.5
Vitamin A	1625	0.4	2737	0.6
Vitamin E	−36	−1.8	−35	−1.4
Vitamin B_1_	−3	−1.4	−6	−2.6 *
Vitamin B_2_	−1	−0.6	−2	−1.5
Vitamin C	−363	−0.9	−197	−0.4
Saturated fatty acids	−26	−0.5	−18	−0.3
Salt	52	2.7 *	43	1.9

Generalized linear mixed model with * *t* > 2 or *t* < −2, with no adjustment factor (Model 1), or adjusted for the preprandial hunger score at breakfast, lunch, and dinner (Model 2).

## Data Availability

The data presented in this study are available on request to the corresponding author. The data are not publicly available due to ethical concerns.

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
