# Peer review of "Inter-Day Variation in the Fasting Plasma Lipopolysaccharide Concentration in the Morning Is Associated with Inter-Day Variation in Appetite in Japanese Males: A Short-Term Cohort Study"

_metabolites, 2023, doi:10.3390/metabo13030395_

Round 1
Reviewer 1 Report
The manuscript „Inter-day variation in the fasting plasma lipopolysaccharide concentration in the morning is associated with inter-day variation in appetite in Japanese males: A short-term cohort study” concerning physiological background of eating habits, an important problem of current world.
It is a well-written text, with meticulous reporting and analysis of data. The criteria of participant selection are explained in detail. The number of participants is low, is this a result of selection only? The participants were selected among company employees (line 98) only, or relatives, friends etc.? Is it possible to comment on the number of company employees and age gradient to assess the pool of possible subjects?
There are some questions that may increase the effect the manuscript has:
LPS is not a unique molecule, various forms are produced by bacteria. Therefore the effects may depend on the combination and “health” of gut flora as well as influx of foreign bacteria, for example from food (yoghurt?) or food supplements. The authors provided extremely thorough information on selection of candidates and the data collected, including log of medicines and supplements (line160). Could the Authors comment on possibility of gut flora differences and the effect on study? Was there a gut microbiome analysis ever considered? The food intake – did the Authors investigated possible foods affecting the quality and quantity of gut bacteria?
In introduction, the effects of LPS are not discussed in relation to bacterial composition of microbiome (ref. 13). In this study, the metabolism of participants was observed individually, however, in the mentioned cases when LPS was injected, the type (origin) of LPS may be important. I would be interested in a comment on this issue.
In line 52, the “In contrast” phase may suggest contrast between negative and positive effects. Was this really the intention of Authors, as LPS appetite stimulating activity is discussed in the following sentences?
The results are presented in an extremely detailed and thorough way, which I find really impressive. Did the Authors explore the effect of vegetable consumption of participants on LPS level? Lower vegetable consumption in at least one day was one of selection conditions. Was there any difference observed? (If I missed this part, I am sorry, the results are overwhelming).
As detection limit for LPS is mentioned (line 116), the value may be added to the results description, with a possible comment on results exceeding this value?
Panel D in figure 1 – is it really necessary as a visual element? The description in the text may be sufficient as there seem to be no noticeable non-correlated points.
Table 3 is extremely long. Is it possible to transform it into a shorter version with the meal data models forming supercoulmns (parallel arrangement)?
The discussion mentions effects of LPS on insulin activity (line 460). Did the authors consider including glucose and insulin monitoring into their study? It is mentioned in future plans (line 464), was there any specific reason this was not included in the current study?
Line 221. Please check this sentence, as “The appetite scores of the participants were recorded in accordance with a procedure described a previous study” – or described in a previous study?
The quality of language is very high, this is the only strange phrase I noticed.
Reviewer 2 Report
This study aimed to evaluate the relationships between Lipopolysaccharide plasma concentration and fat-burning markers or appetite scores in (four) Japanese males. Despite the low number of participants, which is the main limitation, the study design is very well conceived and clearly described. The variables are well defined and M&M description is enough for the replicability of the study. The statistical approach gave consistency to the results as preliminary findings, which need be confirmed or refuted in a further large study with similar design. These issues were clearly mentioned in the manuscript and turn it acceptable for publication. The discussion is balanced regarding the preliminary nature of the findings. I suggest to highlight that the people participation (employees of a single enterprise) in the study was voluntary even an informed consent was obtained (L95).
Reviewer 3 Report
The paper by Nobuo Fuke and colleagues is a communication focusing on the relation between variation in the fasting plasma lipopolysaccharide concentration and inter-day variation in appetite. The topic is interesting and falls within the scopes of the Journal Metabolites.
Abstract is clear and well written. Aiming at improving readability for the general reader of Metabolites, one possible advice would be to avoid reporting Beta and t values at this stage. The reader can find this information in the text.
Introduction is, in general, well written and nice to read. Nevertheless, some parts need rephrasing, such as “Lipopolysaccharide (LPS) is a molecule that is made up”. “Made up” does not have the intended meaning.
The same holds true for Materials and Methods. The authors use several times the expression “due to the fact”. Methods section is very clear and all procedures are well reported.
I would suggest re-organizing Figure 1, which has 3 panels in the first row and only one in the second. A 2 + 2 layout would be better.
Another minor point is related to the overall organization of the paper. Section 4 (Discussion) is interesting and well written, but I do not see the point of two separate sections (5 and 6). Limitations could be overviewed in Discussion section. In this connection, conclusion is rather brief, the authors could consider also including limitations here.
Round 2
Reviewer 1 Report
All the questions were answered, full explanatins were provided. The chanages in the text were explained.
I am impressed by the meticulous work of the Authors.
Reviewer 3 Report
The authors improved the manuscript and addressed all the comments. Thus, it can be approved for publication.